# Interactive-Chain-Prompting: Ambiguity Resolution for Crosslingual Conditional Generation with Interaction

Jonathan Pilault [1 2 3 *]   Xavier Garcia [1]   Arthur Bražinskas [4]   Orhan Firat [1]

## Abstract

Crosslingual conditional generation (e.g., machine translation) has long enjoyed the benefits of scaling. Nonetheless, there are still issues that scale alone may not overcome. For instance, in the absence of additional context, a source query in one language may yield several translation options in another language. Only one translation could be acceptable however, depending on the translator's preferences and goals. Choosing the incorrect option might significantly affect translation usefulness and quality. We propose a novel method *interactive-chain prompting* — a series of question, answering and generation intermediate steps between a *Translator* model and a *User* model — that reduces translations into a list of subproblems addressing ambiguities and then resolving such subproblems before producing the final translated text. To check ambiguity resolution capabilities and evaluate translation quality, we create a dataset exhibiting different linguistic phenomena which lead to ambiguities at inference for four languages. To encourage further exploration in this direction, we **release** all datasets. We note that *interactive-chain prompting*, using eight interactions as exemplars, consistently surpasses prompt-based methods with direct access to background information to resolve ambiguities.

## 1. Introduction

Transformer Language Models (LM, Vaswani et al. 2017) pretrained on large corpora have achieved outstanding results in a variety of NLP benchmarks (Devlin et al., 2019; Brown et al., 2020). Scaling the number of parameters, the

\* Work done while at Google [1]Google Research, Brain Team [2]Mila [3]Polytechnique Montreal [4]Google Research, XGen Team. Correspondence to: Jonathan Pilault <pilaultj@mila.quebec>, Orhan Firat <orhanf@google.com>.

*Interactive Learning with Implicit Human Feedback Workshop at International Conference on Machine Learning (ICML) 2023*, Honolulu, Hawaii, USA. Copyright 2023 by the author(s).

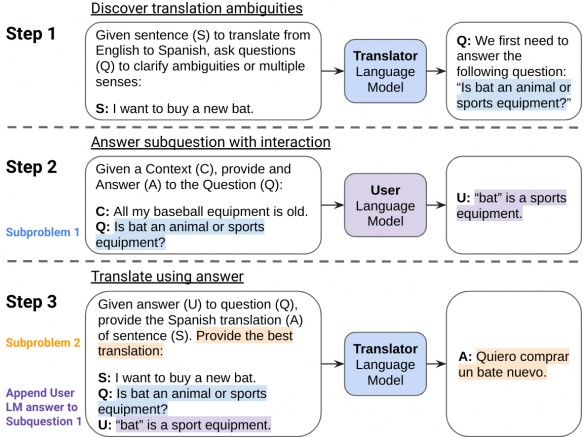

*Figure 1.* Interactive-Chain-Prompting (INTERCPT).

size of the pretraining dataset, and the amount of computing budget gives Language Models better sample efficiency and ability to generalize for many tasks (Kaplan et al., 2020; Brown et al., 2020; Henighan et al., 2020; Hernandez et al., 2021; Lepikhin et al., 2021; Wei et al., 2022a). However, for tasks such as commonsense and symbolic reasoning, where the solution requires multistep computation, or crosslingual conditional generation such as Neural Machine Translation (NMT), where there could be more than one plausible prediction for a given source sequence, scale alone may not be sufficient to achieve high accuracy (Rae et al., 2021; Ghorbani et al., 2022).

Chain-of-thought (Wei et al., 2022b) and least-to-most (Zhou et al., 2022) methods have demonstrated, by prompting a (large-)LM such as PaLM (Chowdhery et al., 2022), that breaking down a task into subproblems that are solved sequentially greatly improves the quality of the final prediction. Such methods demonstrate that producing intermediate sub-results that address specific aspects of a bigger problem significantly improves performance on tasks like arithmetic, math word problems, and symbolic manipulation.While studies have investigated the translation capabilities of PaLM with various prompting strategies (Vilar et al., 2022; Zhang et al., 2023), prompting large and general purpose LMs such as PaLM to identify and solve subproblems in crosslingual conditional generation tasks such as NMT has not yet been fully explored.

Our approach, ***Inter**active-**C**hain-**P**rompting* (INTERCPT), sequentially solves translation subproblems before generating a final translation prediction. As shown in Figure 1, we first detect ambiguities in translation queries, then we resolve these ambiguities via question-answer interactions, and finally we generate translations. INTERCPT departs from other prompt-based techniques that sequentially solve subproblems in two fundamental ways: (1) the subproblems are related but considerably different to the main task and (2) the solutions to subproblems requires interaction with another LLM. In this paper, we will look at how intermediate computation steps and interaction might assist overcome a typical problem in automated systems when a user's ambiguous query leads to a large number of viable and potentially inaccurate answers. In translation, for example, selecting the incorrect prediction has a significant impact on translation quality as illustrated in Fig. 2.

INTERCPT has several advantages. First, the LM is able to identify and ask questions about translation query ambiguities with only a few in-context exemplars and no fine-tuning. This is crucial since large corpora with specific target ambiguities, labels to classify each ambiguity subtypes (i.e. feminine/masculine for gender or formal/informal for formality) and context are not common and are typically low-resource. Then, without readily available context, we rely on the *User* to disambiguate translation queries. In the absence of additional background information or context, there are limited options to solve ambiguities. Interaction with the *User* stands as a logical way to collect clarifying information. This interaction also benefits from multiple computation steps where ambiguity resolution leads to a more precise final prediction. Finally, the question-answer-translation interaction improves transparency and makes it easier to debug translation systems since we can assess the reasoning chain that led to an error (Wu et al., 2022a). For NMT, there are two main questions to consider to make the most of out of intermediate computation steps:

**A) What subproblem are we trying to solve?** Multistep reasoning tasks can often be explicitly decomposed into subproblems: ambiguity detection, disambiguation via Q&A and translation. For NMT, decomposing the translation task is not trivial. We assume in this work that our subproblems are ambiguities which arise when translating. As seen in Fig. 1, the first step in INTERCPT is to discover and resolve the translation ambiguity subproblem. We study five types of ambiguities: polysemous words, pronoun resolution, formality, gender-neutral names and neutral professions. Since datasets that cover multiple translation ambiguities and language pairs while providing context are rare, we create our own datasets (see Table 5 in Section C for an overview of other publicly available datasets).

**B) Where do answers to subquestions come from?** When

Query from the User to Translate from English to French

*Figure 2.* Translation queries with multiple possible predictions. Correctly solving subproblems around ambiguities with **you** and **it** greatly affects the BLEU (Papineni et al., 2002) translation metric.

we apply least-to-most prompting to math word problems for example, the answers to subquestions can often be derived from the problem's text. It is not necessarily the case for NMT where the query may not contain enough context to resolve ambiguities. As seen in Fig. 2, English sentence 'S' does not contain enough information about "you" and "it". The incorrect prediction made by a model leads to large variations in translation quality scores. With more context, the model may have the necessary information to narrow down possible predictions. However, in industrial applications, translation queries are often too short (Badeka, 2016) or additional context is not existent. In this work, we automate interaction between a *PaLM Translator* model, that detects ambiguities, asks clarifying questions and translates, and a *PaLM User* model, that has access to context and answers questions. Both models engage in a multiturn dialog to zero-in on a narrower set of predictions. We argue that a type of question-answer interaction with a "user" is necessary to resolve ambiguous queries, especially when a user (1) is unfamiliar with the main task and may not possess the in-domain knowledge to choose from many model prediction options; (2) knows how to answer simple pointed questions about a query but may not be able or willing to decide and add appropriate context on the fly.

This work marks Large-LM's potential to leverage a few in-context examples, to provide natural language answers and deliver results closer to a user's intent.

## 2. Interactive-Chain-Prompting (INTERCPT)

When interacting with a model, a user may have some well-conceived query in mind that is inadvertently under-specified. For example, a monolingual English speaker may be unaware that the pronoun "you" in a sentence can lead to formal or informal constructs in other languages. The

model may therefore not receive additional information on the level of formality needed to adequately translate the text by this particular user.

A human translator, when asked to translate queries with "you", may want to first probe the user's latent context about the query by asking clarifying questions. In doing so, the human translator can use the answers to better align the translation to a User's request and context. Our method endows language models (LMs) with the ability to generate a similar chain of interactions between a Translator LM and a User LM as seen in Fig. 1. In real applications, it is expected that a human replaces the User LM. INTERCPT uses in-context exemplars to resolve ambiguities before completing the crosslingual conditional generation task that the model is originally asked to do.

It consists of a three step reasoning chain (See Fig. 1) with demonstrations that remain constant for each input query:

1. **The first step is for identifying ambiguities.** The prompt in this step contains the exemplars, showing multiple queries to translate and questions about each query's ambiguities. During inference, the *Translator* LM uses the prompt to generate a pointed question that identifies the specific ambiguity.

2. **The second step is for resolving ambiguities.** The prompt in this step contains exemplars answering the question to the ambiguity subproblems in step one. The *User* LM answers each question using additional information from the provided context. In real life applications, we assume that a real user has similar background information about the text to be translated.

3. **The third step is for translating.** Generated questions and answers are appended to the prompt in step 1 before the final translation is produced. Constant prompts in this step demonstrate how to translate in the specified target language using only details provided by the *User* LM and no-context. During inference, the *Translator* LM uses the prompt to generate the translation.

## 3. Ambiguity MT Datasets (AMBIGMT)

In this section, we introduce AMBIGMT, a dataset that covers four language pairs, for translations from English into French (en-fr), German (en-de), Spanish (en-es) or Japanese (en-ja) — 18 sub-tasks in total. The parallel translation corpora contain five types of ambiguities: "it" resolution, formality, polysemy, gender[1] neutral names, neutral professions. Unless otherwise specified, all datasets include 1000 diverse samples for each {en-fr, en-de, en-es, en-ja} language pair extracted from Opensubtitles corpora (Lison

---

[1]Please note that due to the lack of large translation corpora with various genders and the complexity in creating non-binary gender datasets, our data is limited to feminine and masculine.

---

*Table 1.* AMBIGMT data examples for each ambiguity for target language $x$. $\Delta$ B is the BLEU performance drop from 100 if the highlighted ambiguity is resolved incorrectly.

| Dataset | *en* Query | Context | *x* Target | $\Delta$ B |
|---|---|---|---|---|
| **"it" resolution** | He has read **it** to me so many times that I've learnt **it** by heart. | - I remember when the **postcard** came, Ernesto was so pleased. - He said: "Look what my Rosetta has written to me". | Me **la** sé de memoria de tanto **leerla**. | -44 |
| **Polysemy** | **head** | If you don't feel well, **head** home. | 先 | -100 |
| **Formality** | The closer **you** can get to him, the better. | - I'm aware of the risks, **Master Jedi**, but I know **you** can regain Clovis' trust. | Plus **vous serez** proche de lui, mieux **cela** sera. | -58 |
| **Gender neutral names** | **Blair** should be wrapping up **[pr]** breakfast with Beatrice. | - I have **her** doorman on retainer. - There's a fine line between surveillance and stalking. | Blair sollte **ihr** Frühstück mit Beatrice haben. | -40 |
| **Neutral professions** | **[pr]** worked previously as a businesswoman, accountant, and bank executive. | **Margaret** Mhango Mwanakatwe is a Zambian politician [...]. **She** was the director for business development [...] | Previamente, trabajó como **empresaria**, **contadora** y **ejecutiva** bancaria. | -70 |

& Tiedemann, 2016). In Section C of the Appendix, we provide more details on datasets and describe the heuristics to identify ambiguities in each language.

**"it" resolution** data contains English sentences where the pronoun "it" does not clearly refer to a noun within the query. In English, the pronoun "it" is a singular, neuter and impersonal pronoun. In other languages, "it" may translate into gender specific pronouns (either feminine or masculine) or get dropped entirely from the sentence. The choice depends on what the pronoun refers to. To correctly translate, the model must first determine what "it" is. In the first example of Table 1 where the target language $x$ is Spanish, knowing that "it" is a postcard, or *una tarjeta postal* in Spanish, disambiguates gender in the translation. While the gender affects two words in the target sentence, the wrong gender choice is not only qualitatively inappropriate but also decreases quality metrics (44 BLEU score drop from 100).

**Polysemy** is a dataset that contains words that have multiple meanings and the query is insufficiently informative to zero-in on a specific sense. The context uses the word within a sentence to provide the necessary background information. In the second example of Table 1 where the target language $x$ is Japanese, the context shows that "head" is a verb. In conjunction with the noun "home", we disambiguate "head" as "to move in the direction of". In the absence of such context, "head" has various senses such as "upper part of the body", "side of a coin", "end of a hammer or tool", "a toilet on a boat", "to hit the ball with the head", "to lead".

**Formality** is a dataset where English queries contain the pronoun "you". In the target languages studied, "you" can be formal or informal. As seen in the third example of table 1 where the target language $x$ is French, the speaker addresses the listener "you" as "Master Jedi" in the context,

a title implying a formal style of politeness. The formality is ambiguous without the context and may impact the generated translation quality. Indeed, an incorrect choice in formality level changes "vous serez" to "tu seras" and "cela" to "ça", decreasing BLEU scores by 58 points from 100.

**Gender Neutral Names** data includes queries where the name is gender neutral and ambiguous. The fourth example in Table 1 shows a query where the name "Blair" is gender neutral. In this dataset, we replace gendered pronouns in the English query by the token *[pr]* to remove hints about gender type. From the context, the speaker employs "her" and we can infer that a feminine pronoun "ihr" should be used in the translated German text.

**Neutral Professions** has 600 unique samples for two language pairs. This dataset is derived from the Translated Wikipedia Biographies dataset[2] that covers {en-de, en-es}. In this dataset, the gender of typically gender-neutral professional designations is not clear from the English query alone. In the fifth example of Table 1, the context provides additional hints that the query is talking about "Margeret", also designated by the feminine pronoun "she". Resolving gender allows the model to correctly translate the list of professions in the query and potentially limiting the 70 points drop in BLEU scores from 100.

## 4. Experimental Setup and Results

In this section, we present the main cross-lingual generation results of INTERCPT for formality, "it" resolution and polysemy ambiguity resolution subtasks.

**Setup.** We use PaLM (Chowdhery et al., 2022), a 540B-parameter decoder-only LM pretrained on primarily English-centric data with ∼20% of the data obtained from non-parallel multilingual corpora. The *generalist* prompt template is composed of two formality, three polysemy and three "it" resolution exemplars. All prompt-based methods are 8-shot with the same source sentences $S$ to translate and corresponding translated sentences $A$ in the target language. Each target language has its own prompt template since $A$ differs with every language. The simulated LM user is based on a single English-only 8-shot prompt template for all target languages. Example 4.1 shows the structure of an the LM user prompt exemplars for polysemy. A complete overview of all prompts and exemplars used in experiments can be found in Sections D.1 for the User LM and Sections D.2 for the generalist Translator LM.

**Example 4.1.** *Given a Context (C), provide an Answer (A) to the Question (Q):*
*S: about*
*C: About 2% of the households are enumerated using the*

canvasser method.
*Q: Is "about" an adverb that means approximately, near or a preposition that means regarding, over, surrounding?*
*A: "about" means approximately.*

**Baselines.** Our main baselines were chosen to compare the cross-lingual generation abilities of large multipurpose LMs given interaction, context or no additional information. Please note that, to the best of our knowledge, there are no other baselines that (1) explore large multipurpose LM's capability on contextualized (or interactive) multilingual translation; (2) do not require finetuning on large datasets.

LLMwCXT, our strongest baseline, is the only PaLM-based prompt method that benefits from having *all of the background information required* to resolve ambiguities. LLMwCXT has a prompt with exemplars formulated as the one in example 4.2. In the example, references to **you** and **it** are directly accessible in context $C$.

LLMNOEXTRA is a PaLM-based prompt method that does *not* receive additional information to resolve ambiguities. This baseline is not only of interest for performance comparison and to evaluate model bias but also it can provide insights on the usefulness of additional background information to disambiguate queries. The structure of a LLMNOEXTRA exemplar is similar to example 4.2 without the context $C$. The model must translate the source sentence $S$ in the target language without knowing details about "it" or the level of formality to employ for "you".

GTRANSLATE is a commercially available multilingual and multipurpose baseline queried using the Google Cloud Translation API[3]. This baseline allows us to set performance expectations that LLMNOEXTRA model should reach.

**Example 4.2.** *Given context (C), Translate (S) from English to French:*
*S: Are **you** sure that **it** is pretty?*
*C: She was trying on a new **hat**. Looking at herself in the mirror, she asked her **friend Isabelle**.*
*A: **Es-tu** certaine qu'**il** est **beau**?*

**Metrics.** Our evaluation includes the standard BLEU and BLEURT (Sellam et al., 2020) automatic translation quality metrics as well as additional measures that assess specific ambiguity resolution capabilities. For formality, we use a rule-based classifier to quantify generated sentence formality levels (F-Acc) in the target language. We discuss details of the heuristics in Appendix E. Note that the formality classifier is based on the formality data creation scripts that allowed us to automatically identify formal and informal sentences in the source corpus. For "it resolution", we found that the PaLM 62B-parameter model was surprisingly accurate at identifying translated sentence genders (G-Acc). As seen in Table 7 of Appendix E, PaLM 62B achieves

---

[2]https://ai.googleblog.com/2021/06/a-dataset-for-studying-gender-bias-in.html

[3]https://translate.google.ca/

*Table 2.* Translation results using an 8-shot generalist template that contains exemplars for formality, "it" resolution and polysemy ambiguity types. F-Acc = formality accuracy, G-Acc = gender accuracy, B@n = BLEURT@n. BLEU and BLEURT results for INTERCPT labelled with † are significantly better than all other systems based on pair-wise significance testing (Koehn, 2004) with p = 0.05.

| Lang. Pairs | Method | Formality | | | "it" resolution | | | Polysemy | | | |
| --- | --- | --- | --- | --- | --- | --- | --- | --- | --- | --- | --- |
| | | BLEU | BLEURT | F-Acc. | BLEU | BLEURT | G-Acc. | Hit@3 | Hit@10 | B@3 | B@10 |
| **en→es** | INTERCPT | **36.3**† | **77.9**† | **67%** | **33.6**† | **78.9**† | **77%** | **46%** | **48%** | **54.6**† | **56.8**† |
| | LLMwCXT | 34.7 | 77.1 | 64% | 30.8 | 77.2 | 68% | 40% | 46% | 46.9 | 55.1 |
| | LLMNOEXTRA | 34.6 | 77.0 | 62% | 29.6 | 75.9 | 63% | 33% | 40% | 44.9 | 51.0 |
| | GTRANSLATE | 31.4 | 75.3 | 50% | 27.5 | 73.0 | 54% | — | — | — | — |
| **en→fr** | INTERCPT | **39.1**† | **70.6** | **72%** | **35.3**† | **71.7**† | **73%** | **46%** | **48%** | **46.9**† | **48.5**† |
| | LLMwCXT | 36.4 | 69.9 | 65% | 33.5 | 68.4 | 68% | 36% | 40% | 40.1 | 44.7 |
| | LLMNOEXTRA | 35.7 | 69.2 | 63% | 32.3 | 66.7 | 66% | 33% | 37% | 38.1 | 41.8 |
| | GTRANSLATE | 30.7 | 67.4 | 58% | 29.1 | 65.4 | 61% | — | — | — | — |
| **en→de** | INTERCPT | **35.8**† | **75.0** | **69%** | **24.0**† | **76.0** | **75%** | **43%** | **45%** | **45.1**† | **47.6**† |
| | LLMwCXT | 33.6 | 74.6 | 61% | 22.4 | 75.0 | 69% | 35% | 39% | 36.1 | 44.9 |
| | LLMNOEXTRA | 32.5 | 74.4 | 62% | 22.8 | 73.2 | 63% | 32% | 35% | 36.7 | 41.3 |
| | GTRANSLATE | 27.5 | 72.3 | 53% | 22.1 | 73.0 | 59% | — | — | — | — |
| **en→ja** | INTERCPT | **28.6**† | **69.7**† | **67%** | **23.1**† | **72.4**† | **74%** | **41%** | **44%** | **44.7**† | **47.0**† |
| | LLMwCXT | 26.3 | 68.0 | 60% | 21.4 | 70.8 | 67% | 34% | 38% | 35.8 | 43.8 |
| | LLMNOEXTRA | 25.9 | 67.4 | 61% | 21.2 | 70.3 | 61% | 30% | 33% | 34.6 | 37.0 |
| | GTRANSLATE | 23.5 | 66.7 | 50% | 19.9 | 68.6 | 52% | — | — | — | — |

97% and 93% accuracy in classifying samples of generated translations for Spanish and French respectively. For polysemy, we found that exact match metrics did not fully describe the performance of models. Whenever the model generated a synonym of the ground truth, the exact match metric would not consider the prediction correct. The LLM-NOEXTRA polysemy exemplars are a comma-separated list of synonyms. Our hit@$n$ measures whether the ground truth exists in the first $n$ generated words. For example, if the model outputs the list of Spanish words ["aproximadamente", "cerca de", "alrededor de", "casi", "más o menos"], for $n = 3$, hit@3 would return a match for a ground truth target "cerca de" and no-match for a ground truth target "casi". To supplement the hit@$n$ metric, we also report results of a new metric that we call BLEURT@$n$ (B@$n$) which returns the highest BLEURT score of the first $n$ generated word phrases. Since BLEURT captures the non-trivial semantic similarities between words using its contextual representations from BERT, we found that the metric better measures if correct synonyms were generated by the model. Note that we did not report the GTRANSLATE hit@$n$ or B@$n$ numbers since the API only provides single word outputs.

**Discussion.** Our test results for en-es, en-fr, en-de and en-ja are summarized in Table 2. We first notice that INTERCPT surpasses all other baselines. Surprisingly, LLMwCXT, even with all the necessary background to resolve ambiguities, significantly lags behind INTERCPT on F-Acc. for formality, G-Acc. for "it resolution" and both hit@3 and B@3 for polysemy. This results suggests that the multistep computation approach of fist resolving the ambiguity subproblems and then generating text has an advantage over other baselines. BLEU scores are also 2-3 points higher while BLEURT scores are only slightly higher. This suggest that INTERCPT generates sentences syntactically much

closer to the ground truth while conserving the correct semantics.

## 5. Analysis

*Table 3.* Translation results on unseen ambiguity subproblems using the Gender Neutral Names data and with added unseen domain using the Neutral Professions data. INTERCPT results labelled with † are significantly better with p = 0.05.

| Pair | Method | BLEU | BLEURT | G-Acc. |
| --- | --- | --- | --- | --- |
| Gender Neutral Names — unseen ambiguities | | | | |
| **en→es** | INTERCPT | **31.8**† | **74.1**† | **76%** |
| | LLMwCXT | 29.9 | 72.4 | 66% |
| | LLMNOEXTRA | 30.9 | 71.6 | 59% |
| | GTRANSLATE | 27.8 | 66.1 | 56% |
| **en→fr** | INTERCPT | **31.0** | **63.5**† | **71%** |
| | LLMwCXT | 29.5 | 62.6 | 64% |
| | LLMNOEXTRA | 30.0 | 60.9 | 63% |
| | GTRANSLATE | 24.5 | 57.7 | 56% |
| **en→de** | INTERCPT | **17.9**† | **72.2** | **73%** |
| | LLMwCXT | 15.6 | 71.5 | 67% |
| | LLMNOEXTRA | 15.2 | 70.8 | 61% |
| | GTRANSLATE | 17.1 | 67.1 | 55% |
| **en→ja** | INTERCPT | **16.1**† | **70.3**† | **71%** |
| | LLMwCXT | 14.7 | 69.1 | 65% |
| | LLMNOEXTRA | 14.4 | 68.3 | 60% |
| | GTRANSLATE | 14.1 | 66.0 | 54% |
| Neutral Professions — unseen ambiguities + unseen domain | | | | |
| **en→es** | INTERCPT | **37.3** | 75.8 | **70%** |
| | LLMwCXT | 37.1 | **76.1** | 69% |
| | LLMNOEXTRA | 35.5 | 75.7 | 59% |
| | GTRANSLATE | 37.0 | 72.7 | 56% |
| **en→de** | INTERCPT | **14.3** | 70.0 | **68%** |
| | LLMwCXT | 14.0 | **71.9** | 66% |
| | LLMNOEXTRA | 12.2 | 70.0 | 62% |
| | GTRANSLATE | 13.8 | 67.2 | 54% |

Here, we provide key insights on INTERCPT. We show that INTERCPT better generalizes to unseen ambiguities in Sub-

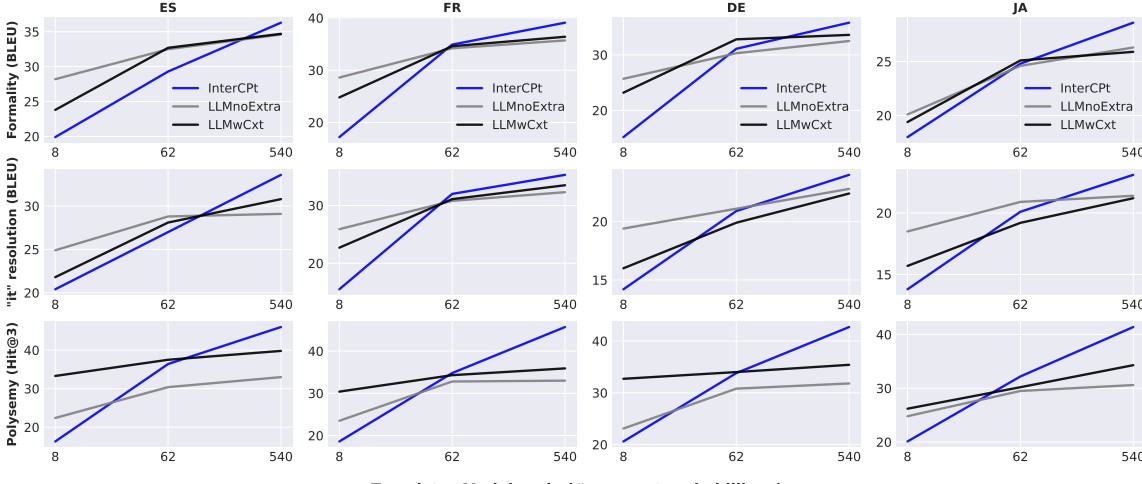

*Figure 3.* INTERCPT enables large LMs to solve ambiguity subproblems in cross-lingual generation. The multistep disambiguate-translate capability is an emergent ability that is reached at higher parameter scales.

section 5.1. In Subsection 5.2, we see that templates with examplars covering many ambiguities (i.e. generalist) performs on par with ones with single ambiguity examplars (i.e. specialist). In Subsection 5.3, we find that interactive translation is an ability that emerges with scale. Subsection 5.4 shows that User LM scale is important, with best scores reached with the 540B-LM model. We study our method's failure modes in Subsection 5.5 showing that where we can improve INTERCPT further. Finally, we provide evidence that interaction helps better mitigate bias in Subsection 5.6.

## 5.1. How does interaction generalize?

In Table 3, we provide translation test results on two held-out datasets that are described in Section 3: (1) Gender Neutral Names and (2) Neutral Professions. We use the same *generalist* prompt template as in Section 4 with examplars that cover only formality, "it" resolution and polysemy. Specifically, our exemplars for both the Translator LM and the User LM do not contain exemplars to resolve the gender for a person's name or profession. We observe that on the Gender Neutral Names dataset INTERCPT performs best on BLEU and BLEURT and is much more able to resolve ambiguities with 6 to 10 points G-Acc improvements over LLMwCXT. On the Neutral Professions data, where test samples are taken from a different domain (Wikipedia biographies instead movie scripts), LLMwCXT and INTERCPT have similar performances. It is possible that LLMwCXT benefits from additional sentences in the context to better determine the style of the output. Nonetheless, INTERCPT provides a 1-2 point increase on G-Acc.

## 5.2. Are specialist better than generalist prompts?

So far, we have studied a *generalist* 8-shot template covering three different types of ambiguities with at most three

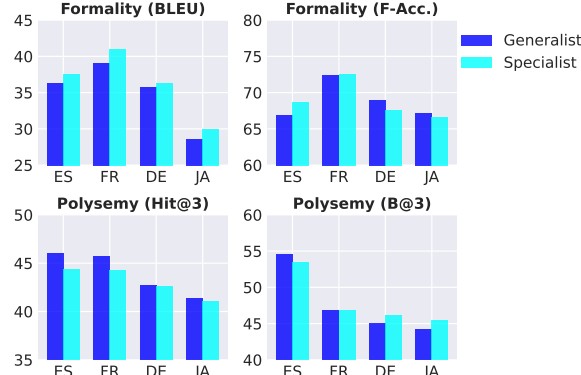

*Figure 4.* Generalist vs Specialist prompt templates for Spanish (ES), French (FR), German (DE) and Japanese (JA) targets.

exemplars per ambiguity. In Fig. 4, we present results of *specialist* template that only covers one type ambiguity at the time (either all formality or all polysemy). Interestingly, specialization does not seem to provide much additional benefit in resolving ambiguities as evidenced by F-Acc, Hit@3 and B@3 results that are on par and often lower than the *generalist* approach. However, the *specialist* template does have a higher BLEU score, implying greater syntactic alignment with the target translation when more ambiguity-specific exemplars are added.

## 5.3. Are interactive generation abilities emergent?

We show in Fig. 3 for each prompt template the effects of scaling PaLM parameters on the performance of formality, "it" resolution and polysemy for Spanish (ES), French (FR), German (DE) and Japanese (JA) target languages. Please note that while we vary the parameter count (8B, 62B and 540B) of the Translator LM, the User LM is a 540B parameters PaLM model for all experiments. The plots provide

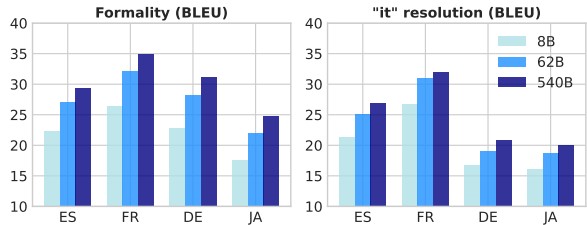

*Figure 5.* Scaling Simulated User LM improves the performance of a 62B Translator LM model.

interesting insights. First, at the 8B parameter scale, LLM-NOEXTRA performs best across all languages for Formality and "it" resolution across all language pairs. Neither context or interaction seem to provide benefits to translation. Second, at the 62B parameter scale, the LLMWCXT and INTERCPT methods have on par performances. Context or interaction in this case are only clearly beneficial for polysemy. Third, the PaLM 540B parameter INTERCPT outpaces other prompt-based methods across language pairs and ambiguity subproblems. At this stage, baselines scaling trend decelerates, with *scaling curves flattening*, compared to INTERCPT. It shows that INTERCPT is an emergent ability of model scale (Wei et al., 2022a). We conjecture that the emergent behavior of INTERCPT is due to a better ability to ask questions and incorporate answers before generating final prediction.

### 5.4. How important is User LM parameter scale?

While the User LM allows us to automate the evaluation of interactivity for cross-lingual generation, it is not clear if the quality of the answer to the Translator LM questions impact performance. We hypothesize that a larger User LM model would provide higher quality answers and allow the Translator LM to better generate translated text. Fig. 5 shows that, when the Translator LM is a 62B PaLM model, a higher parameter User LM improve overall performance. It is therefore possible that answer quality has a significant impact on translation quality and that human-generated answers can further improve overall performance.

### 5.5. When is context better than interaction?

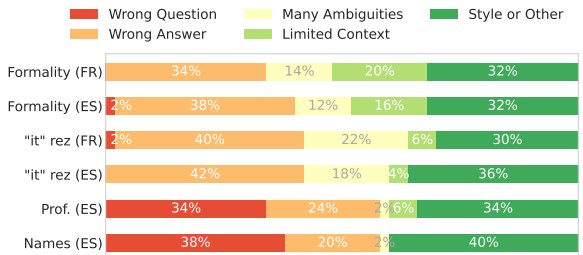

*Figure 6.* Error analysis. rez = "it" resolution, Prof. = Neutral profession, Names = Gender Neutral Names

*Table 4.* Examples of interaction chain errors.

| Error Type | *en* Query (S) and Question (Q) | Sim User Context (C) and Answer (A) | Observation |
|---|---|---|---|
| **Wrong Question** | **S:** But I swear to you it wasn't me. **Q:** What does "it" refer to? | **C:** I just thought that he'd blame me for predicting his death [...]. **A:** "it" is death | **S** can be translated without information on "it" and did ask a question to disambiguate formality. |
| **Wrong Answer** | **S:** Develop it further, Leonard. **Q:** What does "it" refer to? | **C:** -Get me a complete rundown on Miller [...]. -That's a good idea. **A:** "it" is a plan | "plan" is masculine in fr and es. However, "it" refers to "idea", which is feminine in fr and es. |
| **Many Ambiguities** | **S:** If anyone asks, you're a relief worker. **Q:** Who does "you" refer to? | **C:** -Okay, so I'm going to go with you. -White girls don't do runs. **A:** 'informal' since the speaker talking to a friend "Aaron" | The answer is correct however the name Aaron is gender neutral and was resolved incorrectly, impacting "worker" translation. |
| **Limited Context** | **S:** I'll bring it right over. **Q:** What does "it" refer to? | **C:** -You didn't get it? -Really? -Just a second... **A:** "it" is a harp | "harp" is likely wrong. We cannot determine what "it" is from the given context. |

In this section, we provide analysis that describes common areas of improvement for *generalist interactive-chain prompting*. We first isolated test samples for French and Spanish for four ambiguities (formality, "it" resolution, neutral professions and gender neutral names) where the BLEURT scores were less than or equal to LLMWCXT scores. We then randomly sampled 50 interactions and manually analysed the interaction chains (query, question, context, answer, translation). This led us to five types of errors: (1) wrong question, when the Translator LM asked a question not related to the ambiguity; (2) wrong answer, when the User LM did not provide correctly disambiguate; (3) many ambiguities, when the query had multiple unresolved ambiguities or the User LM answer also contained ambiguities; (4) limited context, when the context was not sufficiently informative to resolve ambiguities; (4) style or other, when generated translated text had discernible differences with the ground truth. Fig. 6 shows that the majority of errors are from wrong User LM answers for formality and "it" resolution. This partially confirms our hypothesis in Subsection 5.4. For tasks involving unseen ambiguities, the majority of errors come from the Translator LM with 68% to 78% of sample chains having the wrong question or noticeable differences in generated translated text style or form. We provide examples of interaction chains for each type of error in Table 4.

### 5.6. Can interaction help solve NLG bias issues?

Gender bias is a common phenomenon in automated NMT systems (Borkan et al., 2019; Stanovsky et al., 2019; Saunders & Byrne, 2020). Even when there are explicit gender pronouns in the input query or in the context, NMT systems generated text tends to be masculine when translated into languages with grammatical gender (Stanovsky et al., 2019; Saunders & Byrne, 2020; Stafanovičs et al., 2020; Wang et al., 2022). To measure gender bias, all generated trans-

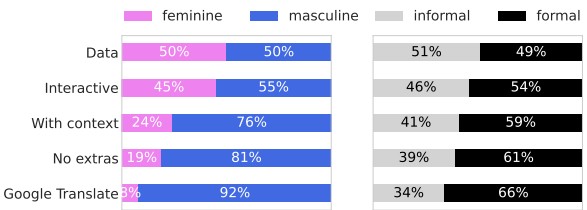

*Figure 7.* Bias in generated translations for French and Spanish on "it" resolution (left) and formality (right).

lations are passed through the gender classifier for the "it" resolution balanced dataset. Similarly, to measure formality bias, generated translations are passed through the formality classifier for the formality balanced dataset. NMT systems can also suffer from formality bias (Rippeth et al., 2022). However, we notice that INTERCPT is much closer to evenly producing masculine and feminine sentences. Our results shows that interactive ambiguity resolution via multistep computation better addresses gender and formality biases.

## 6. Related Works

**Prompting for Cross-Lingual Generation** using Large LMs is a technique that has garnered increasing attention of late. Works on GPT-3 (Vaswani et al., 2017) and PaLM (Chowdhery et al., 2022) show competitive $n$-shot BLEU translation results on WMT. The prompt demonstrations are populated with $n$ random sentence pairs taken from the WMT training corpora and evaluated on the test corpora at inference. Orthogonal to our work, POMP (Vilar et al., 2022) improves upon this PaLM-based prompting technique by explicitly optimizing for the selection of $n$ demonstration sentence pairs and obtaining results competitive with the state-of-the-art. More recent work (Garcia & Firat, 2022) using mT5 (Xue et al., 2021) investigated adding prompt-based natural language specifications to influence translated text properties such as formality level or dialect type. Experiments show that prepending textual artifacts such as "your majesty" to the English query conditions mT5 to generate translations in a formal tone. Our work prompts PaLM with $n$ random translation pair exemplars as well. Different from previous research, we prompt with exemplars to interactively discover background knowledge or clarify ambiguities before translating.

**Interactive Machine Learning** (Ware et al., 2001; Fails & Olsen, 2003; Amershi et al., 2014) is an approach where information is interactively and iteratively supplied to a learning system. In prior interactive translation work, machine interactivity has assisted translators in writing translations by displaying automated word suggestions that update incrementally (Green et al., 2014; Santy et al., 2019). The approach however is limited by drop-down menu options and requires a certain level of sophistication from the

user in the *target language*. Our approach discovers preferences and background knowledge about an input query in the *source language* and more flexibly adapts translations according to a user's natural language response. The interaction is similar to Conversational AI systems where user utterances influence generated outputs. Task or goal oriented conversational AI systems (Konstantinova & Orasan, 2013; Gao et al., 2018; Hussain et al., 2019) are typically deployed to answer knowledge-based questions, seek information or solve basic queries (e.g. making reservations, purchase an item). To the best of our knowledge, our work is the first to explore conversational interaction in cross-lingual generation.

**Resolving ambiguities** by asking for clarifications has been a recent topic of research, for QA and conversational search systems (Lee et al., 2019; Aliannejadi et al., 2019; Zamani et al., 2020; Dhole, 2020; Wang & Li, 2021; Wu et al., 2022b). Departing from such methods, INTERCPT does not produce sentences from a preset list of questions but is generated from a large LM without constrain. Concurrently to our work, Krasheninnikov et al. (2022) explored fine-tuning GPT-3 to generate clarifying questions and provide answers using human generated data from AmbigQA (Min et al., 2020) for open-domain QA. Another GPT-3 model simulates the user and generates answers while conditioned on ground-truth clarification questions. In contrast, our prompt-based method only needs few-shot demonstrations. Further, our simulated user does not rely on ground-truth clarification questions to provide an answer, which could be more realistic for a number of applications (including QA, text simplification, code generation).

## 7. Conclusion

We propose *interactive-chain prompting* (INTERCPT), a prompt-based interactive multistep computation technique that first resolves cross-lingual ambiguities in the input queries and then performs conditional text generation. We have created and released a new datasets that covers five ambiguities: formality, "it" resolution, polysemy, gender neutral names and neutral professions for four different language pairs. Empirical results show that INTERCPT outperforms other prompt-based techniques that have access to all background information and context to directly resolve ambiguities. We find that INTERCPT MT is an emergent property of parameter scale that allows Large LMs to perform interactive generation tasks while other prompt-based techniques exhibit flattening scaling curves. INTERCPT can be considered a step forward more efficiently interacting with machine learning systems.

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

## A. More details on INTERCPT interactive steps

To make link between interaction steps in Figure 1, the process overview in Section 2, the appendix code and templates, we add the following:

Step 1: The Translation LM asks a question on ambiguity using language specific methods in Appendix D.2. It takes as input the English text to Translate *en_text* and outputs the question $Q$. For example, if we want to translate English to Spanish with a generalist template, we can use *spanish_generalist_translator_interactive(...)*.

Step 2: The User LM answers the question $Q$ generated in step 1 using any method in Appendix D.1. It takes as input *en_text* and the context $C$ (ctx in the code) and outputs the answer $U$. For example, we can use *generalist_simulated_user_context(...)*.

Step 3: If no other ambiguity is detected, the Translation LM translates using language specific methods in Appendix D.2. It takes as input the English text to Translate en_text, the question $Q$, and the answer $U$ and outputs the translation $A$.

## B. Link between Chain-of-Thought and Least-to-Most prompting

In this section, we add a few more words on the link between INTERCPT, Chain-of-Thought (CoT) and Least-to-Most (L2M) prompting. CoT performs better than the baseline that has access to the whole information in the problem statement (similar to having context). The behavior is attributed to the sequential solving of subproblems (in our case ambiguity) and a multistep computation (in our case interaction). LLMwCXT has access to more information but does not involve multiple computation steps to solve a subproblem while INTERCPT does

## C. More details on AMBIGMT ambiguity datasets

In this section, we provide additional information on what the datasets contain and how they were created. As mentioned in Section 1, to the best of our knowledge, datasets that cover a large set of ambiguities for multiple language pairs do not exist. We provide an overview of publicly available datasets in Table 5. Upon manual inspection of samples from other public datasets, we found that translation queries were often ($> 50\%$) unambiguous since the translation query contained enough information and did not need to rely on the provided context. We inspected 200 samples from AMBIGMT and found that only 3% of queries did not need context to disambiguate the linguistic phenomena.

*Table 5.* Other MT datasets that contain specific linguistic phenomena and provide context. en = English, de = German, fr = French, ru = Russian, zh = Mandarin Chinese, ja = Japanese.

| Dataset Source | Language Pairs | Linguistic Phenomena | Total Test Data Size |
|---|---|---|---|
| Müller et al. | en→de | (1) "it" pronoun resolution | 12,000 |
| Bawden et al. | en→fr | (1) Anaphora resolution, (2) lexical cohesion | 900 |
| Voita et al. | en→ru | (1) Ellipsis, (2) lexical cohesion | 6,000 |
| Voita et al. | de→en zh→en en→ru | (1) "it" pronoun resolution, (2) lexical cohesion | 6,090 |
| AMBIGMT (**ours**) | en→es en→fr en→de en→ja | (1) "it" pronoun resolution, (2) gender neutral names (3) neutral professions, (4) polysemy, (5) formality | 17,200 |

### C.1. Dataset statistics

We present in Table 6 the data statistics for AMBIGMT. For polysemy, the total senses per word is the number of different definitions or meanings found for a specific source English word. Each ambiguity is well balanced across classes formal/informal or feminine/masculine. The Neutral Professions dataset is derived from the Translated Wikipedia Biographies dataset[4] that only covers {en-es, en-de} language pairs.

---

[4]https://ai.googleblog.com/2021/06/a-dataset-for-studying-gender-bias-in.html

*Table 6.* AMBIGMT data statistics of each type of class and language pair.
Form = formal, Inform = informal, Mas = Masculine, Fem = Feminine, res = resolution, Prof = Profession.

| Language Pair | Total Examples | Polysemy Senses/Word | Formality | | "it" res. | | Neutral Names | | Neutral Prof. | |
|---|---|---|---|---|---|---|---|---|---|---|
| | | | Form. | Inform. | Mas. | Fem. | Mas. | Fem. | Mas. | Fem. |
| **en→es** | 4600 | 3.6 | 49% | 51 % | 50% | 50% | 51% | 49% | 52% | 48% |
| **en→de** | 4600 | 3.1 | 50% | 50 % | 52% | 48% | 50% | 50% | 53% | 47% |
| **en→fr** | 4000 | 3.3 | 49% | 51 % | 50% | 50% | 51% | 49% | — | — |
| **en→ja** | 4000 | 3.0 | 50% | 50 % | 52% | 48% | 53% | 47% | — | — |

## C.2. AMBIGMT data creation tools, process and heuristics

In this section, we present the steps, tools and heuristics used to detect ambiguities. For polysemy, formality, "it" resolution, gender neutral names, we extract the data from OpenSubtitles corpora and neutral professions from Translated Wikipedia Biographies. The source data that was used consists of parallel sentence level pairs. We first detect a sentence that has a specific ambiguity and extract the context by taking three to five preceding English sentences, depending on sentence size. For Polysemy, the context is an English sentence that contains the polysemous word that will be translated. The code and datasets are released **here**.

### C.2.1. POLYSEMY

We provide the following list of steps to create the polysemy dataset for all languages:

1. Extract polysemous words from Wordnet. (Miller, 1994) using the NLTK toolkit (Bird & Loper, 2004)[5].
   - Create a list of English words.
   - Compute the number of definitions per word without counting definitions with synonym overlap.
   - Extract polysemous words ($w_e$) with more than three definitions and a word length greater than four.
2. For each Polysemous English word $w_e$, extract a list $l_x = \{w_{x1}, \ldots, w_{xN}\}$ of possible word translations using the Google Cloud Translation v2 API, where $x \in \{es, fr, de, ja\}$ is the target language.
3. For each Polysemous English word $w_e$ and each target language $x \in \{es, fr, de, ja\}$:
   - Find a sentence that contains the word $w_e$ in the OpenSubtitle dataset.
   - If the parallel sentence contains one of the translated word $w_{xi} \in l_x$ from step 2 and no other translated word, keep the English sentence as context.

### C.2.2. FORMALITY

Each language has specific formality rules. For Japanese, we direct the reader to our public code: `https://anonymous.4open.science/r/interactive_chain_prompting`. We provide the following list of steps to create the formality dataset for Spanish, French and German:

1. Find a sentence that contains "you" or "your" and that has word count less than 20, in the English OpenSubtitle corpus.
2. Select parallel sentences for each target language $x \in \{es, fr, de, ja\}$ that meet the following criteria.
3. If $x == es$, check the following in parallel Spanish sentence (all checks are initialized to FALSE):
   - If all verbs finish by "s", "ste" or "os", then is_verb_informal = TRUE.
   - If any pronouns is "usted", then is_pronoun_formal = TRUE.
   - If any pronouns is in ["tú","tu","te", "vos", "vosotros"], then is_pronoun_informal = TRUE.
   - If any determinants is "su", then is_determinant_formal = TRUE.
   - If any determinants is in ["tu","vosotros", "vosotras"] then is_determinant_informal = TRUE.
   - is_informal = is_verb_informal and is_pronoun_informal and is_determinant_informal.
   - is_formal = is_pronoun_formal and is_determinant_formal.
4. If $x == fr$, check the following in parallel French sentence (all checks are initialized to FALSE):

---

[5]See example in `https://www.nltk.org/howto/wsd.html`

- If any verbs finish by "x", "s" or "ons", then is_verb_informal = TRUE.
- If any verbs finish by "ez", then is_verb_formal = TRUE.
- If one of the pronouns is "vous", then is_pronoun_formal = TRUE.
- If one of the pronouns is "tu", then is_pronoun_informal = TRUE.
- If one of the determinants is in ["vos","votre"], then is_determinant_formal = TRUE.
- If one of the determinants is in ["tes","ton", "ta", "toi"] then is_determinant_informal = TRUE.
- is_informal = is_verb_informal and is_pronoun_informal and is_determinant_informal.
- is_formal = is_verb_formal and is_pronoun_formal and is_determinant_formal.

5. If $x ==$ de, check the following in parallel German sentence (all checks are initialized to FALSE):

- If "!" not in sentence and one of the pronouns is in ["Sie","Ihr", "Ihre", "Ihren", "Ihrem", "Ihrer", "Ihres"], then is_pronoun_formal = TRUE.
- If one of the pronouns is in ["du","dein", "deine", "deinen", "deinem", "deiner", "deines", "dich"], then is_pronoun_formal = TRUE.
- If "!" in sentence one of the pronouns is in ["er","sie", "es", "ihr"], then is_pronoun_formal = TRUE.
- is_informal = is_pronoun_informal.
- is_formal = is_pronoun_formal.

6. Keep samples if is_formal != is_informal, use 'formal' label if is_formal or 'informal' label if is_informal.

7. For each sample, create context by keeping the preceding three to five English sentences, depending if word count is above 20.

### C.2.3. "IT" RESOLUTION

We provide the following list of steps to create the "it" resolution dataset. The steps apply to all languages:

1. For each English sentence in the OpenSubtitle dataset, keep sentences where the word"it" exists.
   - Using a dependency parser, if "it" is expletive[6], skip sample.
   - In the parallel Spanish, French, German or Japanese sentence, if the sentence does not contain a verb and a gendered pronouns, skip sample.
   - Keep gender label.

2. For each sample, create context by keeping the preceding three to five English sentences, depending if word count is above 20.

### C.2.4. GENDER NEUTRAL NAMES

We provide the following list of steps to create the gender neutral names dataset. Please note that for simplicity we used binary genders. Genders beyond female and male will be left for future work. The steps apply to all languages:

1. Compile a list $L_{gnn}$ of gender neutral (unisex) names
   - Collect a list of names with gender statistic such as the percentage of people with the name who identify as female or male[7].
   - Keep the names that are used in approximately equal proportions (unisex) with at least a female or male proportion above 40%.

2. For each gender neutral name $\in L_{gnn}$, find a sentence that contains the name in the English sentence and keep the corresponding parallel sentence in Spanish, French, German or Japanese.
   - If the English sentence has gendered pronouns, skip the sentence if multiple genders are detected.
   - If the English sentence has no gendered pronouns, use a Part-of-Speech tagger[8] on the corresponding parallel sentence in Spanish, French, German or Japanese and skip the sentence if multiple genders are detected.
   - Keep gender label.

3. Replace gendered pronouns with [pr] in the source English sentence to remove simple clues about the name's gender.

---

[6]The spaCy dependency parser can be used to find expletive "it".

[7]Names with gender statistics were compiled and combined using a Japanese names database (Ogihara, 2020) and a English names database that originates from the United States Social Security Administration.

[8]Language specific spaCy models could be used.

4. For each sample, create context by keeping the succeeding three to five English sentences, depending if word count is above 20.

## D. Prompt templates used in experiments

In this section, we discuss the main prompt templates used in experiments. This includes INTERCPT *Translator* generalist and specialist templates to ask questions about ambiguities and exemplars to translate in French, Spanish, German or Japanese. It also includes INTERCPT *User* generalist and specialist templates to answer questions given a context. We also provide the prompt templates for the LLMWCXT experiments where we use context and the same exemplars to translate in French, Spanish, German or Japanese. Please note that we have normalized special characters for simplicity. The German and Japanese templates as well as Spanish and French templates with special characters can be found in our public code and data repository. In the python methods listed below, *en_text* is the input query, *ctx* is the context, *question* is the question from the Translator model and *anwer* is the answer from the User model.

### D.1. INTERCPT Simulated User Prompts

The 8-shot generalist Simulated *User* prompt template is the same for all languages and is provided in code block listing 1.

```python
def generalist_simulated_user_context(en_text, question, ctx):
    """Generalist Simulated user has access to context and answers the question."""

    templated_input =
f"""[web] Given a Context (C), provide an Answer (A) to the Question (Q):

S: about
C: About 2% of the households are enumerated using the canvasser method.
Q: Is "about" an adverb that means approximately, near or a preposition that means
    regarding, over, surrounding?
A: "about" means approximately.

S: rent
C: Many single women cannot live independently because they cannot (afford to) own or rent
    housing
Q: Is "rent" a tenant's regular payment for a property or to pay someone for the use of
    something?
A: "rent" is to pay someone for the use of something.

S: abstract
C: For the international community is not an abstract concept, it consists of us ourselves
    .
Q: Is "abstract" to consider theoretically, to extract something, or a summary, or an
    adjective?
A: "abstract" is an adjective that modifies "concept" in the phrase "abstract concept".

S: What do you mean?
C: Daria, I just think that your field of vision could really be enhanced... - Come on,
    Mom. - It's not my field of vision you want to enhance.
Q: "you" can be neutral, formal, informal. Who does "you" refer to?
A: "you" is 'informal' since the listener is the speaker's "mom", it implies a familiarity
    with the listener "you".

S: This will accelerate your metabolic functions-- help you make the transition.
C: At the very least, get them to hold their fire. - Captain, the transporters are off-
    line. The docking port hasn't been hit yet.
Q: "you" can be neutral, formal, informal. Who does "you" refer to?
A: "you" is 'formal' since "you" refers to a Captain and the speaker will typically use a
    polite form.
```

```
36
37  S: You know where it begins, you never know where it ends...
38  C: Someone once told me we always are where we're supposed to be. - Now I believe it. -
        Life is a journey.
39  Q: "you" can be neutral, formal, informal. Who does "you" refer to in (S)?
40  A: "you" is \'neutral\' because it is a generic "you" that refers to people in general on
        their journey through life.
41
42
43  S: it is also very pretty.
44  C: Even when it is pouring outside, this umbrella is both practical and elegant.
45  Q: What does "it" refer to?
46  A: "it" is an umbrella.
47
48
49  S: Tell me, why do they have to tilt it?
50  C: -Frog is wrong. - I see here that you play the harp.
51  Q: What does "it" refer to?
52  A: "it" is a harp.
53
54
55  S: {en_text.strip()}
56  C: {ctx.strip()}
57  Q: {question}
58  A:"""
59      return templated_input
```

*Listing 1.* INTERCPT Generalist Simulated User Prompt Template

The 8-shot *formality* specialist Simulated *User* prompt template is the same for all languages and is provided in code block listing 2.

```
1  def formality_simulated_user_context(en_text, question, ctx):
2      """Formality simulated user has access to context and answers the question."""
3
4      templated_input =
5  f"""[web] Given a Context (C), provide an Answer (A) to the Question (Q) about Sentence (S
        ):
6
7  S: This is for you, too.
8  C: I'm Freya. - Welcome to Denmark, Mr. Helm. - You always greet people like this? - I'm
        Freya Carlson, your Tourist Bureau contact.
9  Q: "you" can be neutral, formal, informal. Who does "you" refer to in (S)?
10 A: "you" is \'formal\' since "you" refers to a customer or tourist that Freya Carlson is
        greeting with the polite form "Mr.".
11
12
13 S: - i can gladly help you.
14 C: I will go to town to fetch the materials. Once I return, we can repair your majesty's
        royal carriage.
15 Q: "you" can be formal or informal. Who does "you" refer to?
16 A: "you" is \'formal\' since "you" refers to "your majesty".
17
18
19 S: You know what I mean.
20 C: Elizabeth, will you bring the binoculars? - [Elizabeth] Mm, the stench is horrible. [
        John] Here, take a hold of this. - [Elizabeth] Is it dead?
21 Q: "you" can be neutral, formal, informal. Who does "you" refer to in (S)?
22 A: "you" is \'informal\' since the listener "John" has familiarity with the speaker and
        uses the first name "Elizabeth".
23
24
25 S: You think you can make it through that kind of stuff, you think you can make it through
        anything.
26 C: Well, transitions are hard. - Been together ever since college. - Been through a lot. -
```

```
            You know, us coming out to her family, and her brother dying.
27 Q: "you" can be neutral, formal, informal. Who does "you" refer to in (S)?
28 A: "you" is \'neutral\' because it is a generic "you" that refers to people in general
        going through a difficult moment.
29
30
31 S: You can imagine the princess-sized tantrum that followed.
32 Q: "you" can be neutral, formal, informal. Who does "you" refer to in (S)?
33 C: This is the bike that I learned to ride on. - I just didn't know my mom kept it. - It
        used to have these training wheels on the back with lights that would flash every time
         you pedaled. - Then one day, my mom took them off and said it was time to be a big
        girl.
34 A: "you" is \'informal\' since the speaker is talking about a funny childhood memory which
         implies a familiarity with the listener "you".
35
36
37 S: Can I just say, it's been an absolute pleasure to finally meet you?
38 C: Generations of Daleks just woke up very cross, and they're coming up the pipes. - Or to
         put it another way... bye! - Doctor, you must help me.
39 Q: "you" can be neutral, formal, informal. Who does "you" refer to in (S)?
40 A: "you" is \'formal\' since "you" refers to a "Doctor" that the speaker just met.
41
42
43 S: You know where it begins, you never know where it ends...
44 C: Someone once told me we always are where we're supposed to be. - Now I believe it. -
        Life is a journey.
45 Q: "you" can be neutral, formal, informal. Who does "you" refer to in (S)?
46 A: "you" is \'neutral\' because it is a generic "you" that refers to people in general on
        their journey through life.
47
48
49 S: City policemen questioned many of you this week.
50 C: Lying on his belly, he was carried home on a makeshift stretcher. - Next Sunday, after
        the service, the Baron asked the pastor to let him speak.
51 Q: "you" can be neutral, formal, informal. Who does \"you\" refer to in (S)?
52 A: "you" is \'formal\' since the speaker directly addresses several people or "many of you
        ", the plural form of "you".
53
54
55 S: {en_text.strip()}
56 C: {ctx.strip()}
57 Q: {question}
58 A: """
59     return templated_input
```

*Listing 2.* INTERCPT **Formality** Specialist Simulated User Prompt Template

The 8-shot *polysemy* specialist Simulated *User* prompt template is the same for all languages and is provided in code block listing 3.

```
1 def polysemy_simulated_user_context(en_text, question, ctx):
2     """Polysemy simulated user has access to context and answers the question."""
3
4     templated_input =
5 f"""[web] Given a Context (C), provide an Answer (A) to the Question (Q):
6
7 S: abstract
8 C: For the international community is not an abstract concept, it consists of us ourselves
        .
9 Q: Is "abstract" to consider theoretically, to extract something, or a summary, or an
        adjective?
10 A: "abstract" is an adjective that modifies the word "concept".
11
12
13 S: abstract
```

```
14 C: We need to abstract the data from various studies.
15 Q: Is "abstract" to consider theoretically, to extract something, or a summary, or an
      adjective?
16 A: "abstract" means to extract something.
17
18
19 S: about
20 C: About 2% of the households are enumerated using the canvasser method.
21 Q: Is "about" an adverb that means approximately, near or a preposition that means
      regarding, over, surrounding?
22 A: "about" means approximately.
23
24
25 S: about
26 C: The story is about soldier returning home after the war.
27 Q: Is "about" an adverb that means approximately, near or a preposition that means
      regarding, over, surrounding?
28 A: "about" means regarding.
29
30
31 S: bank
32 C: The online banking application does not work. I tried a few times and I could not
      transfer the funds. I went to the bank.
33 Q: Is "bank" a financial institution, the edge of a river, a set or series of similar
      things or the cushion of a pool?
34 A: "bank" is a financial institution.
35
36
37 S: rent
38 C: Many single women cannot live independently because they cannot (afford to) own or rent
      housing
39 Q: Is "rent" a tenant's regular payment for a property or to pay someone for the use of
      something?
40 A: "rent" is to pay someone for the use of something.
41
42
43 S: bat
44 C: The bat flew over the forest and back to its cave.
45 Q: Is "bat" an animal or a sports equipment?
46 A: "bat" is an animal.
47
48
49 C: {ctx}
50 Q: {question}
51 A: """
52     return templated_input
```

*Listing 3.* INTERCPT **Polysemy** Specialist Simulated User Prompt Template

### D.2. INTERCPT Generalist Prompt Templates for each target language

The 8-shot *Spanish* generalist *Translator* prompt template is the same for all test ambiguity data and is provided in code block listing 4.

```
1 def spanish_generalist_translator_interactive(en_text, question=None, answer=None):
2     """Translation model asks questions and uses answers to translate"""
3     if answer == None:
4         #  Ask questions
5         instructions = "[web] Given sentence 'S' to translate to Spanish, ask clarifying
    questions 'Q' to clarify ambiguities or multiple senses:"
6     else:
7         #  Translate given answer
8         instructions = "[web] Given answer 'U' to question 'Q', provide the Spanish
    translation 'A' of sentence 'S'. Provide the best answer:"
```

```
 9
10      templated_input =
11 """
12
13 S: about
14 Q: Is "about" an adverb that means approximately, near or a preposition that means
       regarding, over, surrounding?%s
15
16
17 S: rent
18 Q: Is "rent" a tenant's regular payment for a property or to pay someone for the use of
       something?%s
19
20
21 S: abstract
22 Q: Is "abstract" to consider theoretically, to extract something, or a summary, or an
       adjective?%s
23
24
25 S: You think if I get contacts I'll suddenly turn into the homecoming queen.
26 Q: "you" can be neutral, formal, informal. Who does "you" refer to?%s
27
28
29 S: This will accelerate your metabolic functions-- help you make the transition.
30 Q: "you" can be neutral, formal, informal. Who does "you" refer to?%s
31
32
33 S: They could wait 'till you're on the beach, then cut loose, or start firing right away.
34 Q: "you" can be neutral, formal, informal. Who does "you" refer to?%s
35
36
37 S: can't they just build it on an angle?
38 Q: What does "it" refer to?%s
39
40
41 S: It is also very pretty.
42 Q: What does "it" refer to?%s
43
44
45 """
46     if answer is None:
47         templated_input = templated_input % ('', '', '', '', '', '', '', '')
48         templated_input = f"{instructions}\n" + templated_input + f"S: {en_text}\nQ:"
49     else:
50         templated_input = templated_input % (
51             '\nU: "about" means approximately.\nA: aproximadamente, cerca de, alrededor de
       , casi, mas o menos',
52             '\nU: "rent" is to pay someone for the use of something.\nA: alquilar,
       arrendar, rentar',
53             '\nU: "abstract" is an adjective that modifies "concept" in the phrase "
       abstract concept".\nA: abstraccion, abstracto',
54             '\nU: "you" is \'informal\' since the listener is the speaker's "mom", it
       implies a familiarity with the listener "you".\nA: Tu piensas que si uso lentes de
       contacto de repente me convertire en la nueva reina del colegio.',
55             '\nU: "you" is \'formal\' since "you" refers to a Captain and the speaker will
        typically use a polite form.\nA: Esto acelerara sus funciones metabolicas. Lo ayudara
        a hacer la transicion.',
56             '\nU: "you" is \'neutral\' because it is a generic "you" that refers to people
        in general and not someone specific.\nA: Podian aguardar a que uno estuviera en la
       playa y atacar o comenzar a disparar.',
57             '\nU: "it" is a harp.\nA: no pueden hacerla en angulo?',
58             '\nU: "it" is an umbrella.\nA: Es muy bonita tambien.',
59         )
60     templated_input = f"{instructions}\n" + templated_input + f"S: {en_text}\nQ: {question
       }\nU: {answer}\nA: "
```

```
61        return templated_input
```

*Listing 4.* INTERCPT **Spanish** Generalist Translator Prompt Template

The 8-shot *French* generalist *Translator* prompt template is the same for all test ambiguity data and is provided in code block listing 5.

```
1  def french_generalist_translator_interactive(en_text, question=None, answer=None):
2      """Translation model asks questions and uses answers to translate"""
3      if answer == None:
4          #  Ask questions
5          instructions = "[web] Given sentence 'S' to translate to French, ask clarifying
   questions 'Q' to clarify ambiguities or multiple senses:"
6      else:
7          #  Translate given answer
8          instructions = "[web] Given answer 'U' to question 'Q', provide the French
   translation 'A' of sentence 'S'. Provide the best answer:"
9
10     templated_input = """
11
12 S: about
13 Q: Is "about" an adverb that means approximately, near or a preposition that means
   regarding, over, surrounding?%s
14
15
16 S: rent
17 Q: Is "rent" a tenant's regular payment for a property or to pay someone for the use of
   something?%s
18
19
20 S: abstract
21 Q: Is "abstract" to consider theoretically, to extract something, or a summary, or an
   adjective?%s
22
23
24 S: You know where it begins, you never know where it ends...
25 Q: "you" can be neutral, formal, informal. Who does "you" refer to?%s
26
27
28 S: This is for you, too.
29 Q: "you" can be neutral, formal, informal. Who does "you" refer to?%s
30
31
32 S: You know where it begins, you never know where it ends...
33 Q: "you" can be neutral, formal, informal. Who does "you" refer to?%s
34
35
36 S: I'll help you find it before [pr] does.
37 Q: What does "it" refer to?%s
38
39
40 S: [pr] must have forced it somehow.
41 Q: What does "it" refer to?%s
42
43
44 """
45
46     if answer is None:
47         templated_input = templated_input % ('', '', '', '', '', '', '', '')
48         templated_input = f"{instructions}\n" + templated_input + f"S: {en_text}\nQ:"
49     else:
50         templated_input = templated_input % (
51         '\nU: "about" means approximately.\nA: environ, presque, quelque, a peu pres,
   approximativement',
52         '\nU: "rent" is to pay someone for the use of something.\nA: louer',
```

```
53        '\nU: "abstract" is an adjective that modifies "concept" in the phrase "abstract
      concept".\nA: abstraction, abstrait',
54        '\nU: "you" is \'informal\' since the speaker has familiarity with the listener
      and uses the first name "Jerry".\nA: A qui as-tu parle ?',
55        '\nU: "you" is \'formal\' since "you" refers to a customer or tourist that Freya
      Carlson is greeting with the polite form "Mr.".\nA: Ceci est pour vous.',
56        '\nU: "you" is \'neutral\' because it is a generic "you" that refers to people in
      general going through a difficult moment.\nA: On sait ou cela commence, mais on ne
      sait jamais ou cela se termine...',
57        '\nU: "it" is a key.\nA: Je vous aiderai a la trouver avant elle.',
58        '\nU: "it" is a gate.\nA: Il a du le forcer d\'une maniere ou d\'une autre.',
59        )
60     templated_input = f"{instructions}\n" + templated_input + f"S: {en_text}\nQ: {question
      }\nU: {answer}\nA: "
61     return templated_input
```

*Listing 5.* INTERCPT **French** Generalist Translator Prompt Template

### D.3. INTERCPT Specialist Prompt Templates for each target language

The *Spanish formality* specialist *Translator* prompt template is the same for all test ambiguity data and is provided in code block listing 6.

```
1  def spanish_formality_translator_interactive(en_text, question=None, answer=None):
2      """Translation model asks questions and uses answers to translate"""
3      if answer == None:
4          #  Ask questions
5          instructions = "[web] Given sentence 'S' to translate to Spanish, ask clarifying
      questions 'Q' to clarify ambiguities or multiple senses:"
6      else:
7          #  Translate given answer
8          instructions = "[web] Given answer 'U' to question 'Q', provide the Spanish
      translation 'A' of sentence 'S'. Provide the best answer:"
9
10     templated_input = """
11
12 S: This will accelerate your metabolic functions-- help you make the transition.
13 Q: "you" can be neutral, formal, informal. Who does "you" refer to?%s
14
15
16 S: Poor baby... here's yours!
17 Q: "you" can be neutral, formal, informal. Who does "you" refer to?%s
18
19
20 S: They could wait 'till you're on the beach, then cut loose, or start firing right away.
21 Q: "you" can be neutral, formal, informal. Who does "you" refer to?%s
22
23
24 S: You think if I get contacts I'll suddenly turn into the homecoming queen.
25 Q: "you" can be neutral, formal, informal. Who does "you" refer to?%s
26
27
28 S: For centuries, we have watched you, listened to your radio signals and learned your
      speech and your culture.
29 Q: "you" can be neutral, formal, informal. Who does "you" refer to?%s
30
31
32 S: I never have. I'm not sure you're supposed to.
33 Q: "you" can be neutral, formal, informal. Who does "you" refer to?%s
34
35
36 """
37
38     if answer is None:
```

```
39        templated_input = templated_input % ('', '', '', '', '', '')
40        templated_input = f"{instructions}\n" + templated_input + f"S: {en_text}\nQ:"
41    else:
42        templated_input = templated_input % (
43        '\nU: "you" is \'formal\' since "you" refers to a Captain and the speaker will
      typically use a polite form.\nA: Esto acelerara sus funciones metabolicas. Lo ayudara
      a hacer la transicion.',
44        '\nU: "you" is \'informal\' since the speaker has familiarity with the listener
      and they both use "baby" and "buddy" to address each other.\nA: Pobre bebe... aqui
      esta el tuyo!',
45        '\nU: "you" is \'neutral\' because it is a generic "you" that refers to people in
      general and not someone specific.\nA: Podian aguardar a que uno estuviera en la playa
      y atacar o comenzar a disparar.',
46        '\nU: "you" is \'informal\' since the listener is the speaker\'s "mom", it implies
       a familiarity with the listener "you".\nA: Tu piensas que si uso lentes de contacto
      de repente me convertire en la nueva reina del colegio.',
47        '\nU: "you" is \'formal\' since the speaker addresses people not acquainted with
      or unfamiliar.\nA: Durante siglos, los hemos observado, escuchado sus senales de radio
      . Hemos aprendido su idioma y cultura.',
48        '\nU: "you" is \'neutral\' because it is a generic "you" that refers to people in
      general that have been in this "line of work".\nA: Yo no. No creo que uno deba
      acostumbrarse.'
49        )
50    templated_input = f"{instructions}\n" + templated_input + f"S: {en_text}\nQ: {question
      }\nU: {answer}\nA: "
51    return templated_input
```

*Listing 6.* INTERCPT **Spanish Formality** Specialist Translator Prompt Template

The *Spanish polysemy* specialist *Translator* prompt template is the same for all test ambiguity data and is provided in code block listing 7. Please note that the instructions for the translation step is different than the generalist or the formality specialist template.

```
1  def spanish_polysemy_translator_interactive(en_text, question=None, answer=None):
2      """Translation model asks questions and uses answers to translate"""
3      if answer == None:
4          #  Ask questions
5          instructions = "[web] Given an English word 'S' to translate to Spanish, to
      clarify ambiguities and understand multiple senses ask questions 'Q':"
6      else:
7          #  Translate given answer
8          instructions = "[web] Given answer 'U' to question 'Q', Translate word 'S' into
      Spanish and provide unique and non-repeating synonyms in 'A':"
9
10     templated_input = """
11
12 S: abstract
13 Q: Is "abstract" to consider theoretically, to extract something, or a summary, or an
      adjective?%s
14
15
16 S: abstract
17 Q: Is "abstract" to consider theoretically, to extract something, or a summary, or an
      adjective?%s
18
19
20 S: about
21 Q: Is "about" an adverb that means approximately, near or a preposition that means
      regarding, over, surrounding?%s
22
23
24 S: bank
25 Q: Is "bank" to tilt sideways, or a financial institution, the edge of a river, a set or
      series of similar things or the cushion of a pool?%s
```

```
26
27
28  S: rent
29  Q: Is "rent" a tenant's regular payment for a property or to pay someone for the use of
        something?%s
30
31
32  """
33
34      if answer is None:
35          templated_input = templated_input % ('', '', '', '', '')
36          templated_input = f"{instructions}\n" + templated_input + f"S: {en_text}\nQ: "
37      else:
38          templated_input = templated_input % (
39          '\nU: "abstract" is an adjective that modifies "concept" in the phrase "abstract
        concept".\nA: abstraccion, abstracto',
40          '\nU: "abstract" means to extract something.\nA: abstraer',
41          '\nU: "about" means approximately.\nA: aproximadamente, cerca de, alrededor de,
        casi, mas o menos',
42          '\nU: "bank" is a financial institution.\nA: banco',
43          '\nU: "rent" is to pay someone for the use of something.\nA: alquilar, arrendar,
        rentar'
44          )
45      templated_input = f"{instructions}\n" + templated_input + f"S: {en_text}\nQ: {question
        }\nU: {answer}\nA: "
46      return templated_input
```

*Listing 7.* INTERCPT **Spanish Polysemy** Specialist Translator Prompt Template

The *French formality* specialist *Translator* prompt template is the same for all test ambiguity data and is provided in code block listing 8.

```
1   def french_formality_translator_interactive(en_text, question=None, answer=None):
2       """Translation model asks questions and uses answers to translate"""
3       if answer == None:
4           #  Ask questions
5           instructions = "[web] Given sentence 'S' to translate to French, ask clarifying
        questions 'Q' to clarify ambiguities or multiple senses:"
6       else:
7           #  Translate given answer
8           instructions = "[web] Given answer 'U' to question 'Q', provide the French
        translation 'A' of sentence 'S'. Provide the best answer:"
9
10      templated_input = """
11
12  S: This is for you, too.
13  Q: "you" can be neutral, formal, informal. Who does "you" refer to?%s
14
15
16  S: To whom have you been talking?
17  Q: "you" can be neutral, formal, informal. Who does "you" refer to?%s
18
19
20  S: You know where it begins, you never know where it ends...
21  Q: "you" can be neutral, formal, informal. Who does "you" refer to?%s
22
23
24  S: You can imagine the princess-sized tantrum that followed.
25  Q: "you" can be neutral, formal, informal. Who does "you" refer to?%s
26
27
28  S: City policemen questioned many of you this week.
29  Q: "you" can be neutral, formal, informal. Who does "you" refer to?%s
30
31
```

```
32 S: You think you can make it through that kind of stuff, you think you can make it through
      anything.
33 Q: "you" can be neutral, formal, informal. Who does "you" refer to?%s
34
35
36 """
37
38     if answer is None:
39         templated_input = templated_input % ('', '', '', '', '', '')
40         templated_input = f"{instructions}\n" + templated_input + f"S: {en_text}\nQ:"
41     else:
42         templated_input = templated_input % (
43         '\nU: \nA: Ceci est pour vous.',
44         '\nU: \nA: A qui as-tu parle ?',
45         '\nU: \nA: On sait ou cela commence, mais on ne sait jamais ou cela se termine...'
      ,
46         '\nU: \nA: Tu peux imaginer la colere de princesse qui a suivi.',
47         '\nU: \nA: Les gendarmes sont venus interroger nombre d\'entre vous.',
48         '\nU: \nA: On pense que quand on arrive a traverser ce genre de chose, on peut
      traverser n\'importe quoi.'
49         )
50     templated_input = f"{instructions}\n" + templated_input + f"S: {en_text}\nQ: {question
      }\nU: {answer}\nA: "
51     return templated_input
```

*Listing 8.* INTERCPT **French Formality** Specialist Translator Prompt Template

The *French polysemy* specialist *Translator* prompt template is the same for all test ambiguity data and is provided in code block listing 9. Please note that the instructions for the translation step is different than the generalist or the formality specialist template.

```
1 def french_polysemy_translator_interactive(en_text, question=None, answer=None):
2     """Translation model asks questions and uses answers to translate"""
3     if answer == None:
4         #  Ask questions
5         instructions = "[web] Given an English word 'S' to translate to French, to clarify
      ambiguities and understand multiple senses ask questions 'Q':"
6     else:
7         #  Translate given answer
8         instructions = "[web] Given answer 'U' to question 'Q', Translate word 'S' into
      French and provide unique and non-repeating synonyms in 'A':"
9
10    templated_input = """
11
12 S: abstract
13 Q: Is "abstract" to consider theoretically, to extract something, or a summary, or an
      adjective?%s
14
15
16 S: abstract
17 Q: Is "abstract" to consider theoretically, to extract something, or a summary, or an
      adjective?%s
18
19
20 S: about
21 Q: Is "about" an adverb that means approximately, near or a preposition that means
      regarding, over, surrounding?%s
22
23
24 S: bank
25 Q: Is "bank" to tilt sideways, or a financial institution, the edge of a river, a set or
      series of similar things or the cushion of a pool?%s
26
27
```

```
28  S: rent
29  Q: Is "rent" a tenant's regular payment for a property or to pay someone for the use of
        something?%s
30
31
32  """
33
34      if answer is None:
35          templated_input = templated_input % ('', '', '', '', '')
36          templated_input = f"{instructions}\n" + templated_input + f"S: {en_text}\nQ: "
37      else:
38          templated_input = templated_input % (
39          '\nU: "abstract" is an adjective that modifies "concept" in the phrase "abstract
        concept".\nA: abstraction, abstrait',
40          '\nU: "abstract" means to extract something.\nA: abstraire, extraire',
41          '\nU: "about" means approximately.\nA: environ, presque, quelque, a peu pres,
        approximativement',
42          '\nU: "bank" is a financial institution.\nA: banque',
43          '\nU: "rent" is to pay someone for the use of something.\nA: louer'
44          )
45      templated_input = f"{instructions}\n" + templated_input + f"S: {en_text}\nQ: {question
        }\nU: {answer}\nA: "
46      return templated_input
```

*Listing 9.* INTERCPT **French Polysemy** Specialist Translator Prompt Template

## D.4. LLMWCXT Generalist Prompt Templates for each target language

The 8-shot PaLM-with Context *Spanish* generalist prompt template is the same for all test ambiguity data and is provided in code block listing 10.

```
1   def spanish_baseline_generalist_translator_context(en_text, ctx):
2       """Translation model uses context to translate."""
3
4       templated_input = f"""[web] Given context 'C', Translate 'T' from English to Spanish:
5
6   C: About 2% of the households are enumerated using the canvasser method.
7   T: about
8   A: aproximadamente, cerca de, alrededor de, casi, mas o menos
9
10
11  C: Many single women cannot live independently because they cannot (afford to) own or rent
        housing
12  T: rent
13  A: alquilar, arrendar, rentar
14
15
16  C: For the international community is not an abstract concept, it consists of us ourselves
        .
17  T: abstract
18  A: abstraccion, abstracto
19
20
21  C: Daria, I just think that your field of vision could really be enhanced... – Come on,
        Mom. – It's not my field of vision you want to enhance. – What do you mean?
22  T: You think if I get contacts I'll suddenly turn into the homecoming queen.
23  A: Tu piensas que si uso lentes de contacto de repente me convertire en la nueva reina del
        colegio.
24
25
26  C: At the very least, get them to hold their fire. – Captain, the transporters are off-
        line. – The docking port hasn't been hit yet.
27  T: This will accelerate your metabolic functions–– help you make the transition.
28  A: Esto acelerara sus funciones metabolicas. Lo ayudara a hacer la transicion
```

```
29
30
31  C: Some of the guys got a little sick. - They were scared; I was scared. - I don't think
        we had any reason to be otherwise.
32  T: They could wait 'till you're on the beach, then cut loose, or start firing right away.
33  A: Podian aguardar a que uno estuviera en la playa y atacar o comenzar a disparar.
34
35
36  C: Even when it is pouring outside, this umbrella is both practical and elegant.
37  T: It is also very pretty.
38  A: Es muy bonita tambien.
39
40
41  C: -Frog is wrong. - I see here that you play the harp. - Tell me, why do they have to
        tilt it?
42  T: can't they just build it on an angle?
43  A: no pueden hacerla en angulo?
44
45
46  C: {ctx}
47  T: {en_text}
48  A:"""
49      return templated_input
```

*Listing 10.* LLMWCXT **Spanish** Generalist Prompt Template

The 8-shot PaLM-with Context *French* generalist prompt template is the same for all test ambiguity data and is provided in code block listing 11.

```
1   def french_baseline_generalist_translator_context(en_text, ctx):
2       """Translation model uses context to translate."""
3
4       templated_input = f"""[web] Given context 'C', Translate 'T' from English to French:
5
6   C: About 2% of the households are enumerated using the canvasser method.
7   T: about
8   A: environ, presque, quelque, a peu pres, approximativement
9
10
11  C: Many single women cannot live independently because they cannot (afford to) own or rent
        housing
12  T: rent
13  A: louer
14
15
16  C: For the international community is not an abstract concept, it consists of us ourselves
        .
17  T: abstract
18  A: abstraction, abstrait
19
20
21  C: I believe! - -Who else knows? - -I don't know. - Jerry, names! - I don't want to dance!
22  T: To whom have you been talking?
23  A: A qui as-tu parle ?
24
25
26  C: I'm Freya. - Welcome to Denmark, Mr. Helm. - You always greet people like this? - I'm
        Freya Carlson, your Tourist Bureau contact. - These are for you. Street maps, places
        of interest.
27  T: This is for you, too.
28  A: Ceci est pour vous.
29
30
31  C: It's like the city's changed her. - Well, transitions are hard. - Been together ever
        since college. - Been through a lot. - You know, us coming out to her family, and her
```

```
        brother dying.
32 T: You know where it begins, you never know where it ends...
33 A: On sait ou cela commence, mais on ne sait jamais ou cela se termine...
34
35
36 C: Even when it is pouring outside, this umbrella is both practical and elegant.
37 T: it is also very pretty.
38 A: il est aussi tres beau.
39
40
41 C: Okay, you don't smash the cherry on that. Just plop it in at the end.
42 T: Try to keep it in the top of the glass.
43 A: Essaie de la garder dans le haut du verre.
44
45
46 C: {ctx}
47 T: {en_text}
48 A:"""
49     return templated_input
```

*Listing 11.* LLMwCXT **French** Generalist Prompt Template

## D.5. LLMwCXT Specialist Prompt Templates for each target language

The PaLM-with Context *Spanish Formality* specialist prompt template is the same for all test ambiguity data and is provided in code block listing 12.

```
1 def spanish_baseline_formality_translator_context(en_text, ctx):
2     """Translation model uses context to translate."""
3
4     templated_input = f"""[web] Given context 'C', Translate 'T' from English to Spanish:
5
6 C: At the very least, get them to hold their fire. - Captain, the transporters are off-
      line. - The docking port hasn't been hit yet.
7 T: This will accelerate your metabolic functions-- help you make the transition.
8 A: Esto acelerara sus funciones metabolicas. Lo ayudara a hacer la transicion.
9
10 C: Who? - Me! - I think I've got a cold. - "Hey buddy, give me a Magic Hug will you!" -
      Magic Hug! - And me? - Shut up Swami
11 T: Poor baby... here's yours!
12 A: Pobre bebe... aqui esta el tuyo!
13
14 C: Some of the guys got a little sick. - They were scared; I was scared. - I don't think
      we had any reason to be otherwise.
15 T: They could wait 'till you're on the beach, then cut loose, or start firing right away.
16 A: Podian aguardar a que uno estuviera en la playa y atacar o comenzar a disparar.
17
18 C: Daria, I just think that your field of vision could really be enhanced... - Come on,
      Mom. - It's not my field of vision you want to enhance. - What do you mean?
19 T: You think if I get contacts I'll suddenly turn into the homecoming queen.
20 A: Tu piensas que si uso lentes de contacto de repente me convertire en la nueva reina del
       colegio.
21
22 C: Men of earth, we of the planet Mars give you this warning. - We have known your planet
      earth since the first creature crawled out of the primeval slime of your seas to
      become man.
23 T: For centuries, we have watched you, listened to your radio signals and learned your
      speech and your culture.
24 A: Durante siglos, los hemos observado, escuchado sus senales de radio. Hemos aprendido su
       idioma y cultura.
25
26 C: Pull over here. This is the spot. - I guess you run into a lot of dead bodies in your
      line of work. - You get used to it.
27 T: I never have. I'm not sure you're supposed to.
```

```
28 A: Yo no. No creo que uno deba acostumbrarse.
29
30 C: {ctx}
31 T: {en_text}
32 A:"""
33     return templated_input
```

*Listing 12.* LLMWCXT **Spanish Formality** Specialist Prompt Template

The PaLM-with Context *Spanish Polysemy* specialist prompt template is the same for all test ambiguity data and is provided in code block listing 13.

```
1 def spanish_baseline_polysemy_translator_context(en_text, ctx):
2     """Translation model uses context to translate."""
3
4     templated_input = f"""[web] Given context 'C', Translate 'T' from English to Spanish:
5
6
7 C: Many single women cannot live independently because they cannot (afford to) own or rent
      housing
8 T: rent
9 A: alquilar, arrendar, rentar
10
11
12 C: We need to abstract the data from various studies.
13 T: abstract
14 A: abstraer
15
16
17 C: About 2% of the households are enumerated using the canvasser method.
18 T: about
19 A: aproximadamente, cerca de, alrededor de, casi, mas o menos
20
21
22 C: The bat flew over the forest and back to its cave.
23 T: bat
24 A: murcielago
25
26
27 C: For the international community is not an abstract concept, it consists of us ourselves
      .
28 T: abstract
29 A: abstraccion, abstracto
30
31
32 C: {ctx}
33 T: {en_text}
34 A:"""
35     return templated_input
```

*Listing 13.* LLMWCXT **Spanish Polysemy** Specialist Prompt Template

The PaLM-with Context *French Formality* specialist prompt template is the same for all test ambiguity data and is provided in code block listing 14.

```
1 def french_baseline_formality_translator_context(en_text, ctx):
2     """Translation model uses context to translate."""
3
4     templated_input = f"""[web] Given context 'C', Translate 'T' from English to French:
5
6 C: I'm Freya. - Welcome to Denmark, Mr. Helm. - You always greet people like this? - I'm
      Freya Carlson, your Tourist Bureau contact. - These are for you. Street maps, places
      of interest.
7 T: This is for you, too.
```

```
 8 A: Ceci est pour vous.
 9
10 C: I believe! - -Who else knows? - -I don't know. - Jerry, names! - I don't want to dance!
11 T: To whom have you been talking?
12 A: A qui as-tu parle ?
13
14 C: It's like the city's changed her. - Well, transitions are hard. - Been together ever
      since college. - Been through a lot. - You know, us coming out to her family, and her
      brother dying.
15 T: You know where it begins, you never know where it ends...
16 A: On sait ou cela commence, mais on ne sait jamais ou cela se termine...
17
18 C: You know, if you're gonna go for a spin, I suggest you get your helmet. - This is the
      bike that I learned to ride on. - I just didn't know my mom kept it. - It used to have
       these training wheels on the back with lights that would flash every time you pedaled
      . - Then one day, my mom took them off and said it was time to be a big girl.
19 T: You can imagine the princess-sized tantrum that followed.
20 A: Tu peux imaginer la colere de princesse qui a suivi.
21
22 C: He was in a state of shock, unable to walk. - Lying on his belly, he was carried home
      on a makeshift stretcher. - Next Sunday, after the service, the Baron asked the pastor
       to let him speak.
23 T: City policemen questioned many of you this week.
24 A: Les gendarmes sont venus interroger nombre d\'entre vous.
25
26 C: I tried to explain... He might have gotten hurt! - I was actually doing him a favour. -
       Someone once told me we always are where we're supposed to be. - Now I believe it. -
      Life is a journey.
27 T: You think you can make it through that kind of stuff, you think you can make it through
       anything.
28 A: On pense que quand on arrive a traverser ce genre de chose, on peut traverser n\'
      importe quoi.
29
30 C: {ctx}
31 T: {en_text}
32 A:"""
33     return templated_input
```

*Listing 14.* LLMwCxt **French Formality** Specialist Prompt Template

The PaLM-with Context *French Polysemy* specialist prompt template is the same for all test ambiguity data and is provided in code block listing 15.

```
 1 def french_baseline_polysemy_translator_context(en_text, ctx):
 2     """Translation model uses context to translate."""
 3
 4     templated_input = f"""[web] Given context 'C', Translate 'T' from English to French:
 5
 6 C: Consequently a strategy has been defined that allows departments to approach its
      implementation in a step-wise manner.
 7 T: approach
 8 A: s'approcher, aborder, contacter, s'adresser
 9
10 C: We need to abstract the data from various studies.
11 T: abstract
12 A: abstraire, extraire
13
14 C: About 2% of the households are enumerated using the canvasser method.
15 T: about
16 A: environ, presque, quelque, a peu pres, approximativement
17
18 C: The bat flew over the forest and back to its cave.
19 T: bat
20 A: chauve-souris
21
```

```
22 C: For the international community is not an abstract concept, it consists of us ourselves
       .
23 T: abstract
24 A: abstraction, abstrait
25
26 C: {ctx}
27 T: {en_text}
28 A:"""
29     return templated_input
```

*Listing 15.* LLMwCxt **French Polysemy** Specialist Prompt Template

## E. More details on gender and formality classifier

The classifiers fall into 2 categories: (1) heuristic based classification, that use the same language rules from section C.2; (2) neural network based classification, using a PaLM 62B model with 8-shot in-demonstration exemplars. We provide below the exemplars that were used to classify gender of French in code block listing 16 and Spanish sentences in code block listing 17. Note that we added exemplars until we had a satisfactory score on our ground truth translated sentence (see Table 7).

```
1 def french_gender_it_classifier_template(en_text, fr_text):
2   """Classify a French sentence as feminine or masculine. 7-shot examples"""
3
4     templated_input =
5 f"""[web] Given French sentence 'F', provide the gender of "it" in English sentence 'T'
      and explain in 'E'. Gender in 'A' must be 'feminine', 'masculine' or 'neutral':
6
7
8 T: lily and marshall decided to sell it for one simple reason.
9 F: lyly et marshall l\'avaient mise en vente pour une seule raison.
10 A: feminine
11 E: It is 'feminine' since "mise" refers to a feminine object.
12
13
14 T: - maybe you need to shake it up.
15 F: - peut-etre qu'il faut le secouer.
16 A: masculine
17 E: It is 'masculine' since "le" refers to a masculine object.
18
19
20 T: i want you to get it for me.
21 F: Je veux que tu me la rapportes.
22 A: feminine
23 E: It is 'feminine' since "la" refers to a feminine object.
24
25
26 T: put it back.
27 F: repose-le.
28 A: masculine
29 E: It is 'masculine' since "le" refers to a masculine object.
30
31
32 T: I'm afraid i won't be able to get it for you.
33 F: Je crains de ne pas pouvoir te l'obtenir.
34 A: neutral
35 E: It is 'neutral' since we cannot determine gender with "l\'" only.
36
37
38 T: that view is even more beautiful when you have someone to share it with.
39 F: elle est encore plus belle si on n'est pas seul.
40 A: feminine
41 E: It is 'feminine' since "it" refers to "view" in English and "vue" in French which is
       feminine.
```

```
42
43
44  T: where's it going?
45  F: ou va-t-il ?
46  A: masculine
47  E: It is 'masculine' since "it" refers to "il" in French which is masculine.
48
49
50  T: {en_text}
51  F: {fr_text}
52  A: """
53      return templated_input
```

*Listing 16.* PaLM prompt template for gender classification of French sentences

```
1  def spanish_gender_it_classifier_template(en_text, es_text):
2      """Classify a Spanish sentence as feminine or masculine. 8-shot examples"""
3
4      templated_input =
5
6  f"""[web] Given Spanish sentence 'F', provide the gender in 'A' and explain in 'E'. Gender
       'A' must be either 'feminine' or 'masculine':
7
8  F: nos habriamos pasado el dia mirandola.
9  A: feminine
10 E: It is 'feminine' since "la" and verb "mirandola" refer to a feminine object.
11
12
13 F: - los peruanos no podian pronunciarlo.
14 A: masculine
15 E: It is 'masculine' since "lo" in verb "pronunciarlo" refers to a masculine object.
16
17
18 F: Quiero decir, me encantaria volver a verlo.
19 A: masculine
20 E: It is 'masculine' since "lo" in verb "verlo" refers to a masculine object.
21
22
23 F: debemos ponerla de vuelta?
24 A: feminine
25 E: It is 'feminine' since "la" in verb "ponerla" refers to a feminine object.
26
27
28 F: -tiene que bebersela o tirarla.
29 A: feminine
30 E: It is 'feminine' since "la" in verbs "bebersela" and "tirarla" refer to a feminine
      object.
31
32
33 F: Guardalo para el proximo barco.
34 A: masculine
35 E: It is 'masculine' since "lo" in verb "Guardalo" refers to a masculine object.
36
37
38 F: \"escuchandola me dan ganas de vivir.\"
39 A: feminine
40 E: It is 'feminine' since "la" in verb "escuchandola" refers to a feminine object.
41
42
43 F: !cambialo al menos!
44 A: masculine
45 E: It is 'masculine' since "lo" in verb "cambialo" refers to a masculine object.
46
47
48 F: {es_text.lower()}
```

```
49 A: """
50    return templated_input
```

*Listing 17.* PaLM prompt template for gender classification of Spanish sentences

We have added the classification heuristics and other classification templates to our public data and code repository.

*Table 7.* PaLM 62B gender classification results on a 100 generated translation samples.

| Spanish | French |
|---------|--------|
| 97% | 93% |

