# OpenReview forum: "Interactive-Chain-Prompting: Ambiguity Resolution for Crosslingual Conditional Generation with Interaction"
_ICML.cc/2023/Workshop/ILHF — ILHF Workshop ICML 2023_

### Official Review · Reviewer_S6AH · 2023-06-11
**Interesting Interactive Translation with LLMs**

**Rating:** 7
**Confidence:** 3

**Review:**

## Summary

This paper presents Interactive-chain-prompting (InterCPt), a technique that sequentially solves translation subproblems before the final translation prediction. Experiments show InterCPt provides better translation accuracy than the traditional prompting approach over multiple baselines. This paper also releases AmbigMT, a dataset with five types of ambiguities covering four languages.

## Strengths

S1: Even though prior works have proposed interactive approaches to improve the translation experience (e.g., [Secondary benefits of feedback and user interaction in machine translation tools](https://aclanthology.org/2001.mtsummit-road.3) (Flournoy & Callison-Burch, MTSummit 2001), and works cited in the paper), the proposed InterCPt method is novel in the context of prompting and LLMs.

S2: The experiment is sound, and the results highlight the benefits of InterCPt in ambiguous translation tasks. The paper also presents a detailed analysis of the experiment results with exciting insights.

S3: The AmbigMT would be a valuable resource for future translation researchers.

S4: The paper is well-written and easy to follow.

## Major weaknesses

W1: InterCPt requires users to answer questions to solve ambiguity in their queries. However, a system can also generate multiple translations in one pass, where each translation corresponds to one interpretation of the query. Then, the system can present all translations and used interpretations (in the source language) to the users. One might argue this approach is more scalable in real applications. I wonder if there is comparison between this approach and InterCPt? A comparative user study may yield interesting findings and highlight the advantages of InterCPt (e.g., users may feel they have more agency when the system asks them questions).

## Minor weaknesses

M1: What does "scaling" mean in the abstract? Larger models?
M2: PaLM model is not publicly available, so there might be reproducibility concerns regarding the study results.

---

### Decision · Program_Chairs · 2023-06-20

Accept